# Corrective Machine Unlearning

**Shashwat Goel**[*]                                        *shashwatnow@gmail.com*
*IIIT Hyderabad*
*ELLIS Institute Tübingen*
*Max Planck Institute for Intelligent Systems*

**Ameya Prabhu**[*]
*University of Oxford*
*Tübingen AI Center*

**Philip Torr**
*University of Oxford*

**Ponnurangam Kumaraguru**
*IIIT Hyderabad*

**Amartya Sanyal**                                        *amsa@di.ku.dk*
*University of Copenhagen*

**Reviewed on OpenReview:** *https://openreview.net/forum?id=v8enu4jP9B*

## Abstract

Machine Learning models increasingly face data integrity challenges due to the use of large-scale training datasets drawn from the Internet. We study what model developers can do if they detect that some data was manipulated or incorrect. Such manipulated data can cause adverse effects including vulnerability to backdoored samples, systemic biases, and reduced accuracy on certain input domains. Realistically, all manipulated training samples cannot be identified, and only a small, representative subset of the affected data can be flagged.

We formalize "Corrective Machine Unlearning" as the problem of mitigating the impact of data affected by unknown manipulations on a trained model, only having identified a subset of the corrupted data. We demonstrate that the problem of corrective unlearning has significantly different requirements from traditional privacy-oriented unlearning. We find most existing unlearning methods, including retraining-from-scratch without the deletion set, require most of the manipulated data to be identified for effective corrective unlearning. However, one approach, Selective Synaptic Dampening, achieves limited success, unlearning adverse effects with just a small portion of the manipulated samples in our setting, which shows encouraging signs for future progress. We hope our work spurs research towards developing better methods for corrective unlearning and offers practitioners a new strategy to handle data integrity challenges arising from web-scale training.

## 1 Introduction

Machine Learning models are increasingly trained on large and diverse datasets, with contributions from numerous users and organizations across millions of web-pages (Schuhmann et al., 2022; Gao et al., 2020). However, data integrity issues significantly impact model performance by introducing systemic biases (Prabhu & Birhane, 2021; Konstantinov & Lampert, 2022) and vulnerabilities (Barreno et al., 2006b; Sanyal et al., 2021; Paleka & Sanyal, 2023). For instance, Carlini et al. (2023) showed that a small manipulated subset of web data can lead to large-scale model poisoning, showing the vulnerability of models to adversarial tactics.

---

[*]Equal contribution

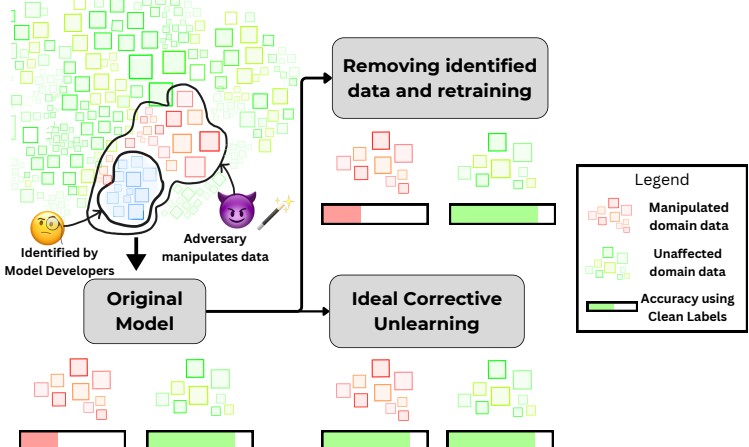

Figure 1: Traditionally, retraining after removing deletion data is considered a gold standard in unlearning, as all samples whose influence is to be removed are assumed to be known. This relies on the retained data not reinforcing the effect to be unlearnt. When developers cannot identify all the manipulated data for corrective unlearning, the retained data can continue to perpetuate the adverse effects of the manipulation. Ideally, corrective unlearning procedures should correct model outputs on the affected domain with access to only a small but representative subset of the manipulated data.

A critical real-world obstacle to eliminating this issue is that model developers can only hope to identify a fraction of this manipulated data due to the size of such large-scale datasets. One practical way for model developers to approach this problem is by using methods for monitoring the integrity of data pipelines (Breck et al., 2019; Wang et al., 2019; Northcutt et al., 2021b) on randomly selected subsets of the whole data. This will identify a small, representative subset of the corrupted samples. Having identified this small subset, the primary goal is to "correct" the negative impacts of corrupted data and their detrimental effect on the model. Due to the high costs already incurred in training and the need to continue uninterrupted service, model developers may wish to update models to remove the influence of the corrupted data instead of abruptly stopping the model's use. We term this problem of removing the influence of manipulated data from a trained model having identified only a small fraction of it, *Corrective Machine Unlearning.* The goal of Corrective Machine Unlearning is to efficiently eliminate the detrimental effects of the corrupted samples even when the precise nature and extent of the manipulation is unknown. In this work, the manipulation is characterised by a small representative subset of the corrupted samples which created the negative impact. Since the manipulation type is assumed to be unknown, this implies that corrective unlearning methods should work across different manipulation types.

Our central claim is that Corrective Unlearning has different underlying requirements from prior work on unlearning motivated by privacy, and thus needs separate attention. Traditional unlearning algorithms (see Nguyen et al. (2022) for a survey) are motivated by the need to cater to user data deletion requests in light of privacy regulations (Council of European Union, 2018; California State Leglisature, 2018; Parliament of Canada, 2018). In contrast, Corrective Unlearning algorithms do not need to obtain privacy guarantees on the "unlearned" data. Instead, corrective unlearning algorithms *must "correct" the effect of the corrupted training data while only having identified a small subset of said data.* We illustrate in Figure 1 why the traditional gold standard in the privacy-oriented unlearning literature is not sufficient for corrective unlearning.

In addition to highlighting the differences of the corrective unlearning setting from traditional unlearning literature, we benchmark popular unlearning procedures (Kurmanji et al., 2023; Goel et al., 2023; Chundawat et al., 2023b; Foster et al., 2023) in this setting. While there are myriad possible ways of constructing manipulations, we choose two manipulations that represent distinct kinds of real-world threats. First, we study a classic poisoning attack (Gu et al., 2019), where an adversary can manipulate both features and labels, making the model misclassify samples that contain a specific trigger pattern, hard for model developers to notice. Second, we study a label-only manipulation called the Interclass Confusion test (Goel et al., 2023), where the adversary incorrectly labels samples between two classes, thereby entangling the model's representations. Such mislabeling can cause systemic biases in model outputs (Prabhu & Birhane, 2021).

Despite being simple manipulations, we find that no unlearning method that we benchmark is able to perform successfully in both scenarios simultaneously. **R**etraining **with**out identified **D**eletion data (referred to as RewoD in this paper) is "Exact Unlearning" when all samples whose influence is to be removed are known. Thus it is considered an inefficient gold standard in traditional unlearning literature, so most popular methods aim to efficiently approximate it (Kurmanji et al., 2023; Goel et al., 2023). However, our experiments show that even knowing 80% of the manipulated data is not enough for RewoD, and consequently other unlearning methods, to remove the adverse effects introduced by manipulating just 1% of the whole training data. Despite this, we find evidence that corrective unlearning with partial manipulation sets known is possible, as the Selective Synaptic Dampening (SSD) (Foster et al., 2023) algorithm is able to remove the effect of BadNet poisoning with just 10% of the manipulated data being identified. There is much scope for progress though, as SSD fails completely in the Interclass Confusion setting, and also leads to a significant drop in overall test accuracy when unlearning poisoned data. We hope our work highlights the need for more effective and rigorous study of unlearning procedures tailored to removing the influence of manipulated data.

## 2 Problem Formulation

In this section, we formalize the requirements of corrective unlearning, highlight the primary challenges, and detail key differences from traditional privacy-oriented unlearning.

### 2.1 Motivation

We first motivate the setting from the adversary's and the model developer's perspectives.

*Adversary's Perspective*: The adversary can manipulate any portion of the input data, including labels in supervised learning scenarios. For example, in poisoning attacks, a trigger is inserted into each manipulated data sample, altering its label to an incorrect one (Han et al., 2022). The goal of the adversary is to inject certain harmful properties into the model learned using this data. While a poisoning adversary injects the backdoor trigger with malicious intent, another adversary may cause systematic mislabelings, resulting in a biased model. We aim to address both types of adversaries.

*Developer's Perspective*: Model developers identify some of the compromised data sources after having already trained a model, either through internal monitoring, defenses, or external information like tip-offs. Since the adversary can apply arbitrary manipulations, the exact manipulation type is unknown to the model developer apriori. As such, *the corrupted data* characterizes the manipulation performed by the adversary. Other ways of characterization, which we are currently out-of-scope for this work include using feedback from model users such as incorrect predictions after deployment or threats identified via red-teaming.

While detecting all manipulated data is challenging, removing its effect given a small subset may be feasible, if the subset is representative of the broader manipulated set. This often occurs in practice when model developers perform expensive but rigorous monitoring on a uniformly sub-sampled subset of the larger dataset. Since this subset is uniformly subsampled, the identified corrupt samples are also a uniformly subsampled subset of the entire set of corrupted samples. The goal of model developers is to remove the adverse effects of the manipulated data from the original model using this small identified representative subset. We formalize this in the next subsection.

### 2.2 Objective of Corrective Unlearning

Let $\mathcal{X}$ be the data domain, $\mathcal{Y}$ the label space, and $\mathcal{P}$ the distribution on $\mathcal{X} \times \mathcal{Y}$. Let $S_{tr} \subset \mathcal{X} \times \mathcal{Y}$ be the training data, and $S_m \subset S_{tr}$ the training samples manipulated by the adversary, either by modifying features, associated labels, or both. Let $\mathcal{D}_m \subset \mathcal{X}$ be the domain where performance is adversely affected when learning using $S_m$. For example, in poisoning, $\mathcal{D}_m$ contains samples with the poison trigger. In Interclass Confusion, $\mathcal{D}_m$ consists of samples from the two affected classes. Clearly, $\mathcal{D}_m$ also contains $S_m$. Finally, let $A$ be the learning algorithm, and $M_o = A(S_{tr})$ be the trained model.

A corrective unlearning algorithm $\mathrm{U_{corr}}$ "corrects" the original model $M_o$ by removing the influence of $S_m$. As mentioned before, we expect only a subset of samples to be identified as manipulated, which we denote

as the deletion set $S_f \subseteq S_m$. Thus, $U_{\text{corr}}$ takes as inputs $M_o, S_{tr}, S_f$ and yields an *unlearned* model $M_u$. Let $\alpha = |S_f|/|S_m|$ be the fraction of the identified manipulated set used for unlearning. For the rest of this work, we assume that $S_f$ is a uniformly sampled subset of $S_m$. The performance of a corrective unlearning algorithm is measured by how well it optimizes the following two objectives for different values of $\alpha$. The faster these two metrics increase as a function of $\alpha$, the better the performance of the unlearning algorithm.

1. **Unlearning**: The primary goal is to unlearn the adverse effect due to the manipulated data $S_m$. We operationalise this as improving the *corrected accuracy on* $\mathcal{D}_m$ while only knowing an $\alpha$ fraction $S_f$ of the full manipulated set:

$$\text{Acc}_{\text{corr}}\left(U_{\text{corr}}, \alpha\right) = \mathbb{E}_{(x,y)\sim\mathcal{P}}\left[\mathbb{I}\{M_u(x) = y\} \mid x \in \mathcal{D}_m\right] \tag{1}$$

   where $M_u = U_{\text{corr}}(M_o, S_{tr}, S_f)$ and $\alpha = |S_f|/|S_m|$.

2. **Utility**: An ideal unlearning algorithm should not harm performance on unrelated samples, *i.e.* outside $\mathcal{D}_m$. Thus, the second goal of the $U_{\text{corr}}$ is to maintain overall accuracy (if not improve it). Using the same notation as above, we operationalize this as the *retain accuracy on* $\mathcal{X} \setminus \mathcal{D}_m$:

$$\text{Acc}_{\text{retain}}\left(U_{\text{corr}}, \alpha\right) = \mathbb{E}_{(x,y)\sim\mathcal{P}}\left[\mathbb{I}\{M_u(x) = y\} \mid x \notin \mathcal{D}_m.\right] \tag{2}$$

Intuitively, the larger the value of $\alpha$, the higher $\text{Acc}_{\text{corr}}$ should be. When the whole manipulated set is known for deletion *i.e.* $\alpha = 1$, retraining without it, gives the best outcome. We dub this $U^*_{\text{corr}} = A(S_{tr} \setminus S_m)$. The performance of an unlearning algorithm $U_{\text{corr}}$ can be measured by how large $\alpha$ needs to be for $\text{Acc}_{\text{corr}}\left(U_{\text{corr}}, \alpha\right)$ to be within some constant fraction (say 0.9) of $\text{Acc}_{\text{corr}}\left(U^*_{\text{corr}}, 1\right)$. We visualize this in our experiments in Section 3.3 and compare different algorithms based on it. Note that, $\text{Acc}_{\text{retain}}$ is relatively less sensitive to $\alpha$ due to the definition of $\mathcal{D}_m$. However, it may not always be possible to accurately identify the entire affected domain $\mathcal{D}_m$. Identifying $\mathcal{D}_m$ can be particularly difficult when the adversary wants to obscure its true target, as in targeted poisoning attacks (Shafahi et al., 2018), or when it wants to affect the accuracy over the entire domain, such as in indiscriminate poisoning attacks (Barreno et al., 2006a; Biggio et al., 2012). Therefore, formulating these objectives for different settings may require more problem-specific attention. Nevertheless, we set the gold standard for $\text{Acc}_{\text{retain}}$ to also be $\text{Acc}_{\text{retain}}\left(U^*_{\text{corr}}, 1\right)$, and in our experiments, this metric turns out to be largely independent of $\alpha$.

### 2.3 Differences from Privacy-Oriented Unlearning

In this section, we discuss the most important distinctions between Corrective Unlearning and Privacy-oriented unlearning. Privacy-oriented unlearning seeks to ensure *retrain indistinguishability* (Sekhari et al., 2021): the unlearning algorithm $U_{\text{corr}}$ aims to produce models indistinguishabile from the models produced by the learning algorithm trained on just the retain set $S_{tr} \setminus S_f$.

**Different Goals (Removal of Incorrect Training Data):** The goal of privacy-oriented unlearning is to remove the influence of untampered but sensitive user data whereas the objective of corrective unlearning is to remove the influence of manipulated samples.

*Implications*: Removing uncorrupted but *private* samples in privacy-oriented unlearning scenarios typically degrades model performance (Golatkar et al., 2020a). This is unavoidable as there is a cost to obtain privacy (Jayaraman & Evans, 2019). Some unlearning procedures even nudge the model output towards a uniform distribution over classes for samples in the forget set (Chundawat et al., 2023b; Li & Ghosh, 2023). However, in corrective unlearning, removing the influence of manipulated samples is expected to improve the model's performance on the affected domain $\mathcal{D}_m$ (i.e. $\text{Acc}_{\text{corr}}$). Intuitively, this may, in turn, improve the learned representations thereby also improving overall accuracy $\text{Acc}_{\text{retain}}$. The underlying cause is that corrective unlearning aims to unlearn corrupt data whereas the focus of privacy-oriented unlearning is removing confidential but not necessarily corrupt data.

**Different Gold-Standards (Retraining without deletion set is not enough):** In privacy-oriented unlearning, the data whose influence is to be removed from the model is specified by user deletion requests. However, as discussed before, when model developers need to identify compromised data, it is unrealistic to

assume all of it will be found. Thus, in corrective unlearning at $\alpha < 1$, the retain set $S_{tr} \setminus S_f$ will continue to contain manipulated data from $S_m \setminus S_f$.

*Implications*: Retraining from scratch on $S_{tr} \setminus S_f$ is the gold standard for privacy-oriented unlearning but it is computationally expensive. Therefore, at its core, privacy-oriented unlearning is a computational problem. However, in corrective unlearning when $\alpha < 1$, as $S_{tr} \setminus S_f$ continues to have manipulated data, unlearning procedures that solely rely on it (Schelter, 2020; Bourtoule et al., 2021; Graves et al., 2021; Goel et al., 2023) perpetuate the adverse effects of the manipulation. This necessitates a search for algorithms beyond computationally efficient approximations of *retraining from scratch*, which ceases to be a gold standard.

Interestingly, this introduces a statistical dimension to the unlearning problem. One possible approach is to use the identified manipulated data to detect other manipulated data. Whether this is possible depends mainly on three parameters: size of the identified manipulated set, dimensionality of the data, and complexity of detecting the manipulated data[1]. For easy-to-detect manipulations in low dimensions and for large identified sets, this problem should be easy. However, existing lower bounds in learning theory (e.g. see Theorem 3.20 in Schapire & Freund (2012)) show that this may well be impossible for small identified sets and hard-to-detect manipulations in high dimensions, which are precisely the realistic conditions we aim to tackle. For example, we show poisoning experiments where the identified subset has 500 samples for 3072-dimensional data and one of the goals of poisons is to be imperceptible. In short, this naturally leads to the following question: How can we *efficiently* remove the detrimental impacts of $S_m$ using a representative, albeit *smaller*, subset $S_f$?

**Different Constraints (No Privacy Requirements):** Finally, in the corrective unlearning context, $S_f$ and $S_m$ does not need to be privatized, setting it apart from privacy-oriented unlearning.

*Implications*: Privacy-oriented unlearning is designed to meet strict privacy standards, necessitating either algorithms with theoretical privacy guarantees (Sekhari et al., 2021; Gupta et al., 2021) akin to those provided by differential privacy, or at least strong performance against privacy auditing on the data to be forgotten $S_f$ (Golatkar et al., 2020a). This further leads to a drop in accuracy as theoretical privacy guarantees are not always tight and lead to underestimation of actual privacy. Such evaluations are orthogonal to the goal of Corrective Unlearning, thereby necessitating the design of new evaluation strategies.

## 3 Experiments

In this work, we run experiments with two types of manipulations that can occur in real-world data collection pipelines, and test whether existing methods can unlearn their adverse effects. We first describe these manipulations, and then our dataset, model architecture, and unlearning setup.

### 3.1 Manipulation Types and Evaluation Benchmarks

As a desiderata for corrective unlearning is dealing with corrupted data, agnostic to manipulation type, a good $U_{corr}$ should correct a broad class of manipulations, including both we test.

**Evaluation of Feature-and-Label Manipulations: Poisoning.** When model developers scrape data from webpages, adversaries can manipulate both the data samples and labels. This leaves models trained on this data vulnerable to backdoor attacks, where the model misbehaves in the presence of trigger features known to the adversary, but unknown to the model developers. Prior work has shown this can be realized in real-world settings such as autonomous driving (Han et al., 2022) and models trained on Wikipedia (Carlini et al., 2023). To model this setting, we use the simple BadNet poisoning attack introduced by Gu et al. (2019). We insert a trigger pattern that makes 0.3% pixels white at bottom-right positions in a subset of $n$ training images, re-labeling each of these images to class zero. Here the affected domain $\mathcal{D}_m$ consists of all samples containing the trigger pattern. We benchmark the ability of unlearning methods to remove the effect of the backdoor after identifying some of the poisoned training data.

**Evaluation of Label-only Manipulations: Interclass Confusion.** When model developers outsource annotations for their training data, adversaries can only manipulate labels. Systematic mislabeling between

---

[1]This is based on classical uniform-convergence based arguments (Alon et al., 1997).

two classes can also occur naturally due to systemic biases in the labelling process, or misinterpretation in annotation guidelines on how to distinguish the classes. While this setting offers lesser power to the adversary, it can still lead to strong mislabeling attacks. Lingam et al. (2024) show that for random label flipping attacks, restricting it to two classes results in the largest drop of accuracy of the Bayes optimal classifier. We thus use the Interclass Confusion (IC) test (Goel et al., 2023), in which, two classes $A$ and $B$ are picked, and half of the samples from both classes are selected, and their label is changed to the other class. Models trained on datasets containing this manipulation are more likely to confuse these classes, i.e. predict samples from $A$ as $B$ and vice-versa. The affected domain $\mathcal{D}_m$ consists of all samples from class $A$ and class $B$. We benchmark the ability of unlearning methods to remove the label confusion after identifying some of the confused data.

## 3.2 Experimental details

We first use the CIFAR10 and CIFAR100 (Krizhevsky et al., 2009) datasets as standard benchmarks in the unlearning literature (Foster et al., 2023; Kurmanji et al., 2023; Chundawat et al., 2023a). We then report poison unlearning results on PCam (Veeling et al., 2018), a binary classification medical imaging dataset, as a potential application. PCam contains histopathologic scans of lymph node sections, with labels indicating the presence of metastatic tissue. For our experiments, we employ the ResNet-9 (Idelbayev, 2018) model for CIFAR10, and the WideResNet-28x10 (Zagoruyko & Komodakis, 2016) model for CIFAR100 and PCam.

In the main paper, we report results for 1% of the training data being manipulated, except for the IC test (which has a weaker adversary than poisoning) on CIFAR-10 where 10% manipulation is required for clear observations. In Appendix B we report results for other manipulation sizes, finding similar unlearning trends even when only 0.2% of the training data is poisoned. In each setting, we vary $\alpha$, the fraction of the manipulated set identified for deletion, from 0.1 to 1.0 at intervals of 0.1. For all results, metrics are computed on the test set containing unseen samples. The mean and standard deviation are reported over 3 seeds. In the interclass confusion evaluation, for CIFAR10, we confuse the Cat and Dog classes, and for CIFAR100, the maple and oak tree classes, to be consistent with the setup of Goel et al. (2023).

**Unlearning Methods.** We select several state-of-the-art unlearning methods to benchmark the effectiveness of current unlearning methods in our setting. Detailed descriptions, along with hyperparameter sweep details for all methods, are provided in Appendix A.2.

*(1) Retrain without Deletion set (RewoD)*: It retrains from scratch on $S_{tr} \setminus S_f$ using the original training algorithm $A$. It is considered an inefficient but gold-standard oracle in the traditional setting of $\alpha = 1$. Many existing methods are relaxations of this algorithm (He et al., 2021; Graves et al., 2021; Goel et al., 2023).

*(2) Catastrophic Forgetting (CF)*: Goel et al. (2023) show that fine-tuning just the final layers of a deep model on $S_{tr} \setminus S_f$ can unlearn label manipulations. We use the strongest version, i.e., fine-tuning all layers.

*(3) Selective Synaptic Dampening (SSD)*: Foster et al. (2023) selectively dampen learned weights with high influence from $S_f$, identified by approximating the Fisher Information Matrix (Martens, 2020). Specifically, they compute each parameter's *relative importance*: the ratio of the forget set and retain set loss sensitivity to the parameter. Parameters with relative importance above a chosen threshold have their value divided by a quantity proportional to the relative importance.

*(4) Knowledge Distillation from a Bad Teacher (BadT)*: Chundawat et al. (2023b) propose making outputs on $S_f$ random by distilling from a randomly initialized network, while distilling the original model on $S_{tr} \setminus S_f$.

*(5) Scalable Remembering and Unlearning Bound (Scrub)*: Kurmanji et al. (2023) propose a method that alternates between two key steps: (1) distillation away from the original model on $S_f$ to remove the influence of the manipulated data, and (2) knowledge preservation using distillation towards the original model combined with a task-specific loss on $S_{tr} \setminus S_f$ to retain useful information.

**How to Select Best Hyperparameters for Unlearning?** We find choosing the hyper-parameters a surprisingly tricky question for corrective unlearning. Selecting the model with the best overall accuracy on a validation set ($\text{Acc}_{\text{retain}}$) may result in low corrected accuracy ($\text{Acc}_{\text{corr}}$) on the domain affected by the manipulation ($\mathcal{D}_m$), especially if it is a small fraction of the overall domain $\mathcal{X}$. Moreover, directly measuring the corrective accuracy may not be possible when the correct labels for the deletion set are not known. We

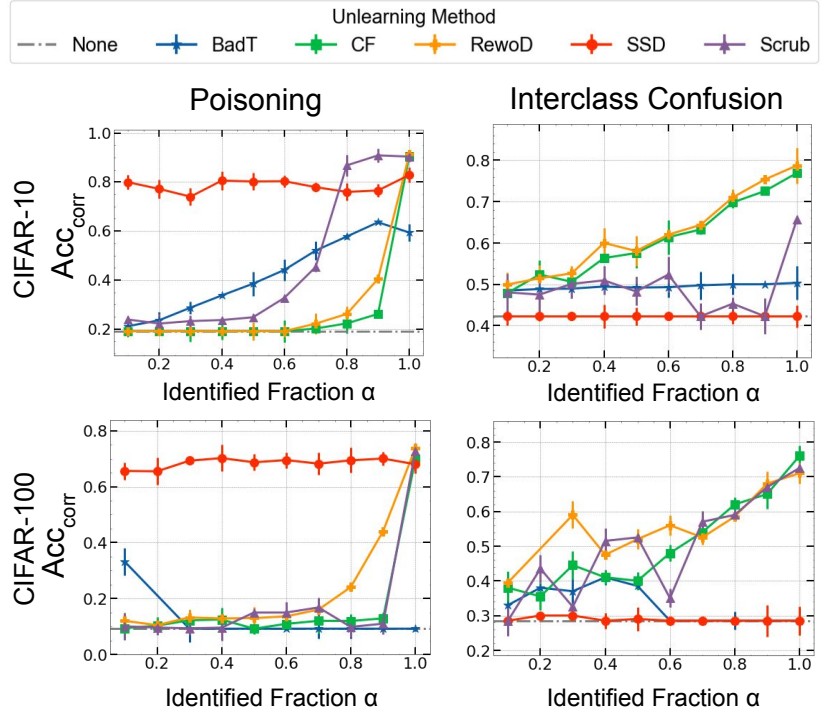

| | CIFAR10 | CIFAR100 |
|---|---|---|
| | **Poisoning** | |
| None | 90.52 | 73.79 |
| **BadT** | $0.29 \pm 0.01$ | $0.02 \pm 0.00$ |
| **CF** | $0.97 \pm 0.06$ | $0.40 \pm 0.05$ |
| **SSD** | $-8.94 \pm 1.02$ | $-3.25 \pm 0.53$ |
| **Scrub** | $0.69 \pm 0.06$ | $0.09 \pm 0.06$ |
| **RewoD** | $0.75 \pm 0.06$ | $0.23 \pm 0.07$ |
| | **Interclass Confusion** | |
| None | 92.36 | 74.35 |
| **BadT** | $0.23 \pm 0.10$ | $-0.04 \pm 0.02$ |
| **CF** | $0.79 \pm 0.14$ | $0.31 \pm 0.07$ |
| **SSD** | $-1.63 \pm 0.95$ | $-5.43 \pm 3.62$ |
| **Scrub** | $-3.27 \pm 1.65$ | $-0.06 \pm 0.05$ |
| **RewoD** | $0.44 \pm 0.15$ | $-0.01 \pm 0.07$ |

Table 1: **Change in retain accuracy ($Acc_{retain}$) after applying different unlearning methods, where higher values are better**. Results are reported as mean $\pm$ stdev over 10 values of the identified fractions $\alpha$ from 0 to 1.0. SSD leads to significant drops in model utility.

Figure 2: **Corrective Accuracy ($Acc_{corr}$) after applying different unlearning procedures**. Each method ("None" represents the original model) is shown across different identified fractions ($\alpha$) of manipulated samples. No method unlearns both the manipulations well when $\leq 80\%$ of the manipulated data is identified, including RewoD which is considered a gold standard at $\alpha = 1$.

propose to use a weighted average between overall accuracy (utility) on a validation set and the fraction of $S_f$ samples where the model's classification changes (unlearning). We weigh them equally for our results.

### 3.3 Experimental Results

Figure 2 shows corrective accuracy trends that quantify the efficacy of different methods at unlearning the adverse effects of manipulations. RewoD is the gold standard when all manipulated samples are known, and indeed it shows the highest accuracy when $|S_f| = |S_m|$. For the IC evaluation, RewoD demonstrates consistent improvement as more deletion set samples are identified. However, for the poisoning evaluation, it shows no improvements when developers detect less than 80% of the manipulated samples, even where only 1% (500 samples) of the training data is manipulated. This highlights the insufficiency of the privacy-based unlearning goal of approximating retraining from scratch on $S_{tr} \setminus S_f$, as the remaining poisoned samples can maintain their adverse effects, even when their number is small (Gu et al., 2019). As a consequence, popular approaches in unlearning literature like Scrub and CF do not perform well in the poisoning setting. Scrub performs better than CF in the poisoning setting, but the opposite is true in the interclass confusion setting. BadT shows poor results in both settings, as randomizing outputs on $S_f$ conflicts with the goal of correcting the model on the affected domain.

On the contrary, SSD recovers $Acc_{corr}$ for poisoning even after identifying just 10% of the manipulated samples, demonstrating manipulations can be unlearnt with a small fraction of manipulated samples identified. However, SSD completely fails for the IC test, providing no improvements over the original model. Further, as shown in Table 1, SSD leads to significant drops in model utility ($Acc_{retain}$), while other unlearning methods maintain utility. Note that we report averaged $Acc_{retain}$ across $\alpha$ as we find it to largely be independent of $\alpha$.

**Conclusion:** Traditional unlearning methods that train on $S_{tr} \setminus S_f$ perform poorly in practical scenarios when not all manipulated samples are known. SSD shows positive results for removing poisons, demonstrating corrective unlearning is possible at $\alpha < 1$, though it fails completely at removing interclass confusion and hurts model utility, leaving scope for improvements.

### 3.4  Why does SSD fail for IC test but work for poisoning?

SSD selects which parameters to dampen using the ratio of the parameter's importance for the forget set versus the retain set. We hypothesise that for poisons, a few parameters are disproportionately more important for the forget set than others, whereas for the IC test, the effect is more distributed among the parameters and no small subset of parameters is particularly important. If this is the case, an approach that dampens high-influence parameters is likely to target a "relevant" parameter in the poisoning setting, effectively reducing its adverse effects, but fail to remove differentially interclass confusion without hurting retain accuracy. To test this hypothesis, we show the distribution of the importance ratio (used to select the weights to dampen) for the two tests in Figure 3. For poisoning, we observe a few outlier weights with disproportionately high values (e.g., 800), whereas for the IC test, the distribution does not show such extreme values (the maximum is less than 200). This empirical observation confirms our hypothesis.

SSD succeeds at removing poisons because it can dampen weights which correlate the poison feature with the wrong label, even with a small portion of the manipulation set known. This is consistent with the well-known strategy of pruning a small subset of weights to mitigate poisons (Wang et al., 2019), as they correlate a specific feature with the mislabel. This may motivate the tractability of mechanistic interpretability-based approaches for unlearning certain types of poisoned data (Elhage et al., 2021).

### 3.5  Application on a Medical Dataset: PCam - Binary Classification

We now examine if our findings generalize to PCam (Veeling et al., 2018), an application-specific dataset used in medical imaging. The only change we expect is that in binary classification tasks, knowing which class is incorrect fully specifies the correct class. Therefore, methods like Scrub which do a form of gradient ascent, moving away from the original manipulated label, are expected to perform well. In Figure 4, we observe that RewoD, CF, and BadT continue to perform poorly until most of the manipulated data is detected, while SSD performs well even at small $\alpha$. As anticipated, the only difference is that Scrub shows significant improvements as moving away from the original model's outputs on the forget set $S_f$ infers the correct labels.

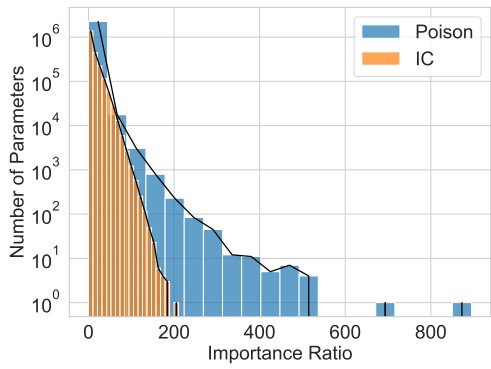

Figure 3: **Histogram of importance ratio values computed for each parameter by SSD**. We find poisoning leads to more outlier values, which supports the hypothesis that poisoning can be removed by dampening outlier parameters unlike Interclass Confusion.

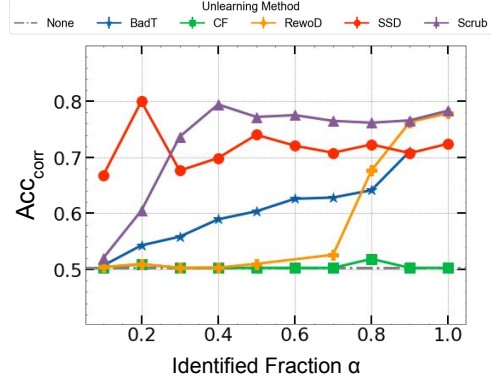

Figure 4: **Corrective Accuracy ($\mathrm{Acc_{corr}}$) for unlearning poisons across different methods on PCam.** Unlearning methods that move away from the original label, such as Scrub, perform well even when less manipulated data is detected, as in this case there is only one other possible label.

## 4  Related Work

**Learning from manipulated Data**: The adverse effects of manipulated training data on machine learning models are well-documented across objectives like fairness (Konstantinov & Lampert, 2022), robustness

(Sanyal et al., 2021), and adversarial reliability (Paleka & Sanyal, 2023; Tian et al., 2022). One line of defense is designing training strategies more robust to these issues, see Song et al. (2022) for a survey on learning with mislabels. However, learning robust models from manipulated data is a hard problem as reduced sensitivity to such minority data populations can harm accuracy and fairness (Sanyal et al., 2022). Unlearning specific samples which are discovered to be manipulated can be a complementary post-training mitigation approach.

**How to detect manipulated data?** A curious reader would wonder if the critical task is detecting the small subset of the data itself. However, detecting manipulated data has long been studied (Brodley & Friedl, 1999), with prior work detailing techniques to discover mislabeled (Pleiss et al., 2020; Northcutt et al., 2021a), biased (Prabhu & Birhane, 2021; Jiang & Nachum, 2020) and poisoned (Chen et al., 2019; Wang et al., 2019) data. We assume the model developers employ such strategies for monitoring their data sources. However, they cannot simply throw away the trained model when manipulated data is found due to expensive retraining costs. Our goal is to study how to cheaply mitigate adverse effects on such models using unlearning.

**Known manipulations**: If the type of manipulation is known, one may employ manipulation-specific mitigation techniques such as poisoning defences against data poisoning attacks (see Goldblum et al. (2022) for a survey) or erasing concepts like artistic style from visual models (Gandikota et al., 2023). If the samples can be corrected through re-annotation, one may also use knowledge editing techniques (Mitchell et al., 2022). We restrict the scope of our work to not knowing the precise manipulation, as corrected data often cannot be obtained, we hope to use unlearning as a broader, panacea procedure across known and unknown data manipulations.

**Unlearning**: Most prior work in designing unlearning procedures is motivated by privacy applications, and aims to achieve *retrain indistinguishability* (Ginart et al., 2019; Golatkar et al., 2020a), that is to create a distribution of unlearnt models indistinguishable from retraining from scratch without the data to be deleted. In Section 2.3 we discuss differences in corrective unlearning desiderata from retrain indistinguishability. "Exact Unlearning" procedures ensure the unlearnt model never sees the data whose influence is to be deleted by design of the training procedure (Bourtoule et al., 2021; Schelter, 2020). The empirical results of RewoD in Section 3.3 show how these approaches may not suffice for corrective unlearning when the full manipulation set is unknown. Moreover, such methods drastically deteriorate in efficiency as the number of samples to delete increase (Warnecke et al., 2021). This has led to "Inexact Unlearning" proposals, and we discuss different paradigms in image classification in Appendix A.2, and use state of the art methods from each paradigm in our experiments. Prior work has also shown that techniques like sparsification (Jia et al., 2023) and regularization (Thudi et al., 2022a) of the original model can improve the performance of unlearning methods. A group of works (Izzo et al., 2021; Wu et al., 2020; Gupta et al., 2021; Neel et al., 2021; Thudi et al., 2022b; Sekhari et al., 2021) also study unlearning procedures on convex or linear models with theoretical guarantees inspired from differential privacy (Dwork, 2006), but in this work we focus on deep models.

Some unlearning works have separately evaluated on both removing backdoors (Sommer et al., 2022; Wei et al., 2023; Liu et al., 2022) and mislabeled samples (Kurmanji et al., 2023; Goel et al., 2023). Indeed, our work proposes neither new methods, nor new evaluations. Instead, we unify these two types of manipulations, to characterize the corrective unlearning as being fundamentally different from the privacy-oriented setting. Kurmanji et al. (2023) suggested different applications of unlearning require different methods. We build on this by formulating the requirements of corrective unlearning, crucially finding retraining cannot be used as a gold standard as all deletion data may not be known, adding a statistical dimension beyond unlearning being just a computational problem. Finally, some prior work has explored unlearning using few or even zero samples, but they either require generative models to unlearn classes (Yoon et al., 2023; Chundawat et al., 2023a) or "counterfactual samples" for unlearning biases (Chen et al., 2024b;a), which is non-trivial to apply for corrective unlearning of manipulations.

# 5 Limitations, Broader Impact, and Conclusion

**Limitations and Future work**  Given the failure of state-of-the-art unlearning methods (including RewoD) at correcting our two simple manipulations, we didn't include more complex manipulations. Ideal corrective unlearning approaches should exhibit robustness against a broad spectrum of manipulation types. Specifically, these methods should withstand adaptive attacks, where the manipulations targeted for unlearning are

crafted with knowledge of the unlearning procedures themselves (Tramer et al., 2020), not just the two manipulations we study. We hope future work will design adaptive adversaries and corresponding unlearning algorithms that can correct manipulations introduced by these adversaries. This provides scope to design stronger evaluation frameworks for corrective unlearning. Apart from manipulating features and labels, adversaries could generate entirely synthetic samples (Zhang et al., 2019; Huang et al., 2020). Although our focus is on supervised image classification, the concept of manipulation and its correction is also relevant in self-supervised learning contexts, such as language modeling (Wallace et al., 2021). An additional complexity could be the presence of false positives, where a clean sample gets identified as manipulated. Corrective unlearning may not always be tractable in all of these settings. Thus, apart from designing new algorithms, we believe Corrective unlearning can inspire a new line of theoretical works, including fundamental questions such as: what are necessary and/or sufficient conditions and size for a 'representative set' of manipulated samples so that corrective unlearning is possible? Can we design procedures to infer more manipulated samples from a representative subset? Can we design algorithms with unlearning guarantees on $\text{Acc}_{\text{corr}}, \text{Acc}_{\text{retain}}$ instead of indistinguishability of distributions? We hope these questions guide future work on corrective unlearning.

**Broader Impact** Corrective unlearning is a post-training strategy designed to mitigate the effects of manipulated or incorrect training data. The societal benefits of this approach are similar to those achieved by defending against manipulations that introduce backdoors or systematic biases, which are largely positive. In some scenarios, such as collective action (Hardt et al., 2023) against harmful models, the ability to manipulate training data could be beneficial. However, defenses against such manipulations might have negative societal effects. These ideas remain speculative, and overall, we believe that removing the effects of manipulated data will have a positive societal impact.

**Conclusion** Overall, we characterize the Corrective Machine Unlearning setting, designed to mitigate the negative effects of manipulated data discovered post-training, such as diminished accuracy on specific parts of the domain. In contrast to the assumption of fully specified user deletion requests in prior work on unlearning, we acknowledge that all the manipulated data samples may not be known. Developers can use a small representative subset of the manipulated samples, which can be collected by careful inspection of a uniform subsample of the training data. We discuss how this leads to significant changes from the traditional unlearning setting, which is primarily designed to address privacy concerns.

Our findings indicate that popular unlearning methods, even retraining on the remaining data (RewoD), fails to enhance accuracy on manipulated domain samples unless nearly all of the manipulated data is identified. A notable exception is SSD, which shows promise by successfully mitigating the effects of the BadNet poison, thus illustrating the feasibility of counteracting manipulated data with only a partial subset identified. However, this method does not generalize across settings, such as addressing Interclass Confusion manipulations. We hope our work spurs the development of stronger corrective unlearning methods to help practitioners deal with data quality issues arising from web-scale training.

## Acknowledgements

SG was funded by an Effective Ventures Foundation (UK) grant as part of the ML Alignment Theory Scholars (MATS) program. AP was funded by Meta AI Grant No. DFR05540. PT thanks the Royal Academy of Engineering and FiveAI for their support. This work is supported in part by a UKRI grant: Turing AI Fellowship EP/W002981/1 and an EPSRC/MURI grant: EP/N019474/1. The authors would like to thank (in alphabetical order): Arvindh Arun, Shyamgopal Karthik, Shashwat Singh, Shiven Sinha, Vishaal Udandarao, Christian Schroeder de Witt for helpful feedback, and Varshita Kolipaka for contributing illustrations.

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

# A  Experimental Setup Details

## A.1  Training Details

Our standard training procedure $A$ is as follows: We train our models for 4000 steps on CIFAR10, PCAM and 6000 steps on CIFAR100. Each step consists of training on a single batch, and we use a batch size of 512 throughout. We use an SGD optimizer with momentum 0.9 and weight decay 5e-4, a linear scheduler with $t_{mult} = 1.25$, and warmup steps as $\frac{1}{100}$ of the total training steps. The same hyperparameters are used during unlearning unless otherwise specified. The setup used for all experiments is a PC with a Intel(R) Xeon(R) E5-2640 2.40 GHz CPU, 128GB RAM and 1 GeForce RTX 2080 GPU.

## A.2  Detailed Description of Unlearning Methods

To benchmark the performance of existing unlearning proposals on corrective unlearning scenarios, we select the strongest unlearning methods across five popular paradigms:

**(1) Retrain without Deletion set (RewoD)**: This paradigm involves retraining parts of the ML system (Bourtoule et al., 2021; Goel et al., 2023; He et al., 2021) that are influenced by $S_f$ from scratch using $S_{tr} \setminus S_f$.

**Method Used:** We benchmark the strongest version, retraining the entire model from scratch on $S_{tr} \setminus S_f$ using the original training algorithm $A$. This is considered an inefficient gold standard in prior work.

**(2) Catastrophic Forgetting (CF)** : Neural Networks suffer from catastrophic-forgetting (French, 1999) - when a model is updated without some previously learnt samples, the model loses knowledge about them. Many unlearning methods perform finetuning on $S_{tr} \setminus S_f$ to achieve unlearning of $S_f$ via catastrophic forgetting, and Goel et al. (2023) showed a weaker efficient version performs well on the IC test.

**Method Used**: We finetune using the original training procedure $A$ for 1000 steps on $S_{tr} \setminus S_f$.

**(3) Modifying learnt parameters with high influence from** $S_f$: This is a training-free class of methods (Golatkar et al., 2020a;b; Peste et al., 2021; Chundawat et al., 2023a) that identifies parameters with information relevant to the forget set using statistics like the Fisher Information Matrix (FIM). It then damages these parameters by adding noise or reducing their magnitude hoping to selectively remove information about $S_f$.

**Method Used**: We benchmark the recently proposed Selective Synaptic Dampening (SSD) method which has shown state of the art results in this paradigm (Foster et al., 2023). We extensively tune the weight selection threshold $\alpha$ and weight dampening constant $\gamma$. We find when $\gamma$ should be tuned relative to $\alpha$ for optimal results. For each datapoint, we pick the best result out of runs with $\alpha = [0.1, 1, 10, 50, 100, 500, 1000, 1e4, 1e5, 1e6]$, $\gamma = [0.1\alpha, 0.5\alpha, \alpha, 5\alpha, 10\alpha]$.

**(4) Pushing** $S_f$ **outputs towards random**: Some unlearning procedures (Graves et al., 2021; Li & Ghosh, 2023; Chundawat et al., 2023b) push the model towards random outputs on the deletion set.

**Method Used**: We benchmark Knowledge Distillation from Bad Teacher (BadT) (Chundawat et al., 2023b), a state of the art method in this paradigm, which simultaneously distills from a randomly initialized neural network on $S_f$, and the original model on the remaining data $S_{tr} \setminus S_f$. We finetune the original model using this procedure for 1000 unlearning steps.

**(5) Alternating between Forgetting and Preservation Steps**:

**Method Used**: Kurmanji et al. (2023) propose SCRUB and show it performs well on unlearning mislabelled samples when all are identified. The method alternates between forget steps and knowledge preservation steps. The forget step involves doing gradient ascent using the task-loss for $S_f$. The knowledge preservation step does knowledge distillation from $M_o$ using $S_{tr} \setminus S_f$ as well as optimizing the task-loss on $S_{tr} \setminus S_f$. We finetune the original model using this procedure for 1000 unlearning steps, out of which the forget step is used only in the first 200 unlearning steps as it is recommended in the paper to run it only in the initial

| Dataset | #Classes | Model | Poisoning $|S_m|/|S_{tr}|$ | IC Test $|S_m|/|S_{tr}|$ |
|---------|----------|-------|----------------------------|---------------------------|
| CIFAR-10 | 10 | ResNet-9 | 0.2%, 1%, 2% | 1%, 5%, 10% |
| CIFAR-100 | 100 | WideResNet-28x10 | 0.2%, 1%, 2% | 0.2%, 0.5%, 1% |

Table 2: Dataset, models and manipulation sizes for the Poisoning and Interclass Confusion (IC) evaluation.

iterations. We use a smaller learning rate (0.0025) as the original value leads to stability issues. We tune the hyperparameter $\alpha$ which controls the trade-off between the distillation loss and the task-loss. For each datapoint, we pick the best result out of runs with $\alpha = [0.001, 0.01, 0.05, 0.1, 0.5, 1, 5, 10]$.

## B  Further Results

We now provide results not included in the main paper due to space considerations. Specifically: (1) We report results across different manipulation set sizes (settings summarized in Table 2), on both unseen samples from the affected domain $\mathcal{D}_m$ (generalization effect of manipulation) and the manipulation set used in training $S_m$ (memorization of manipulation). We do this for both the poisoning and IC evaluation. (2) We report computational efficiency by measuring unlearning time for each method. (3) We report a hyperparameter sensitivity analysis for all methods.

### B.1  Unlearning ($\mathrm{Acc_{corr}}$) results across Manipulation Sizes, and on the Manipulated Set $S_m$

**Setup**  In this section, we vary the manipulation sizes for both poisoning and interclass confusion, to see if the trends are consistent. As the adversary is less powerful (label-only) in IC, we expect IC will need larger manipulation set sizes to show clear trends, while poisoning will work with very small manipulation sets. Due to the changing manipulation sizes, we report the deletion set size on the X-axis, with the total manipulation set size labeled below each subfigure, and also noticeable as the right-most point on the X-axis. In the main paper, we reported unlearning results ($\mathrm{Acc_{corr}}$) on unseen samples from the affect domain $\mathcal{D}_m$, but here we will also investigate the same metric computed on the manipulated samples used in training itself. We thus use the more general term of "clean-label accuracy", i.e. accuracy computed with respect to correct labels, on the Y-axis, which is the same as $\mathrm{Acc_{corr}}$ for results on unseen samples.

**Unlearning Poisoning**  In Figure 5a, we report results on a smaller (0.2%) and bigger (2%) manipulation size than the main paper (1%, also reported here for reference). In both cases, and notably even for the smaller manipulation set, we find consistent results. To measure the removal of mislabelling on poisoned training samples, we report clean-label accuracy on $S_m$ in Figure 5b. The trends across unlearning methods are similar to the ones on unseen samples from the affected domain $\mathcal{D}_m$, though the absolute accuracies after unlearning are higher as expected from training samples in comparison to test set samples. The success of SSD in producing an unlearnt model that successfully classifies the manipulated training data shows how an ideal unlearning algorithm can help re-label the detected data correctly.

**Unlearning Interclass Confusion**  In Figure 6a, we report results on smaller manipulation sizes (0.2%, 0.5% for CIFAR100, and 2%, 5% for CIFAR10) than the main paper (same sizes, i.e. 1% on CIFAR100 and 10% on CIFAR10, also reported here for reference). We see similar trends but with weaker fidelity on the medium setting, but no clear observations on unseen samples in the smallest manipulation size. While the smallest manipulation size (subfigures a, d) for Interclass Confusion did not show significant effects on unseen samples from class $A, B$, Figure 6b shows unlearning methods continue to give wrong predictions on the class $A, B$ samples used for training. This emphasises the need to check unlearnt model outputs on unseen and training samples from the affected domain $\mathcal{D}_m$ separately, especially when the manipulation set is small to have an effect on model behaviour for unseen samples.

### B.2  Computational Efficiency

In Table 3 we report average unlearning times of different unlearning methods. In the case of RewoD and CF, while more efficient relaxations have been proposed (Goel et al., 2023; He et al., 2021; Graves et al., 2021), we retrain from scratch to perform the strongest unlearning, which we still find to be insufficient.

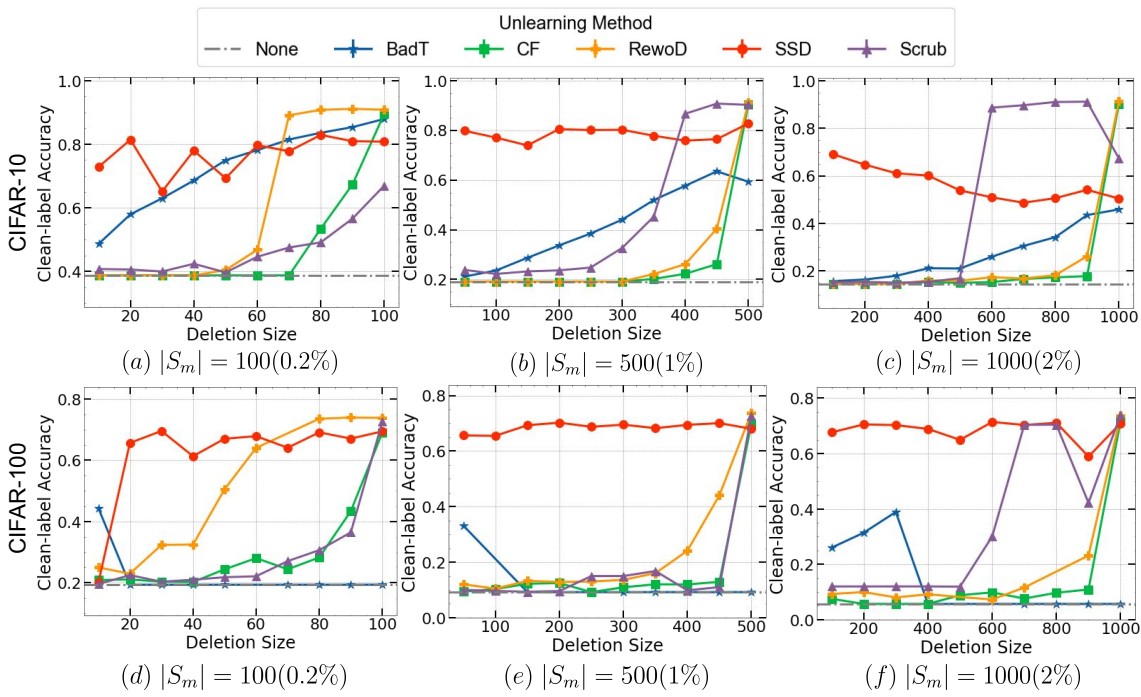

(a) **Clean-label Accuracy on Test Samples with Poison Trigger**. Each method is shown across deletion sizes $|S_f|$ after unlearning ("None" represents the original model). Existing unlearning methods except SSD, including RewoD which is traditionally considered a gold-standard, perform poorly (b), (c), (e), (f) when $\leq 80\%$ of the poisoned data is identified for unlearning, even when just 1% of training data is poisoned as in (b), (e).

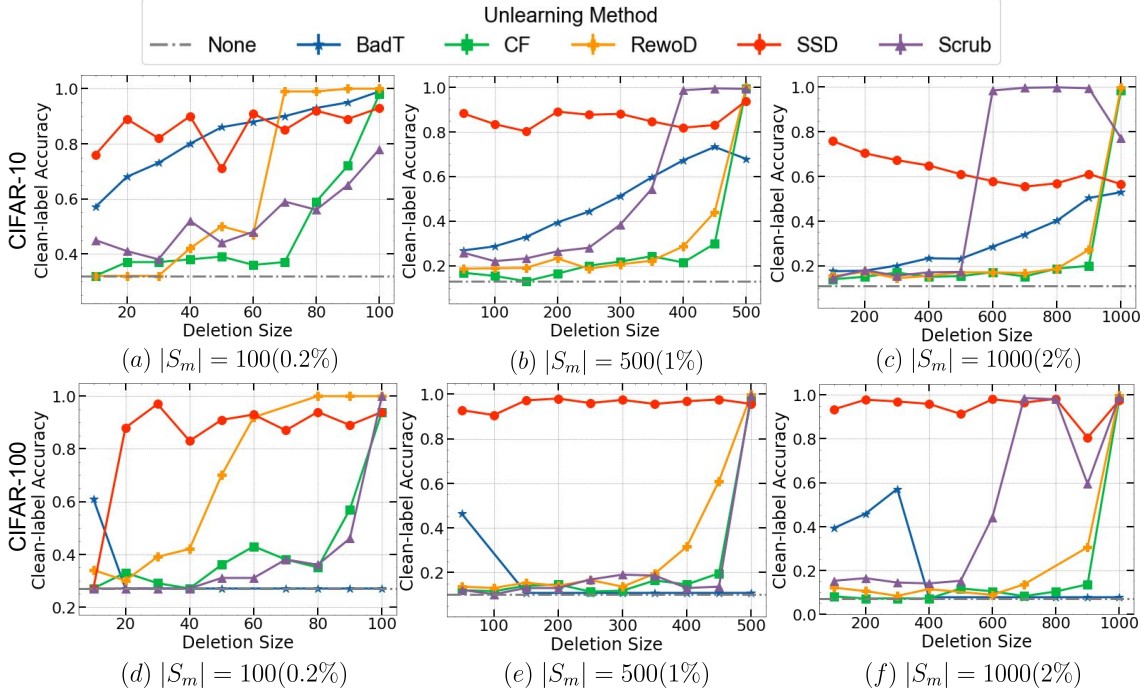

(b) **Clean-label Accuracy on Manipulated Train Samples $S_m$ with Poison Trigger.** Each method is shown across deletion sizes $|S_f|$ after training with adversarial poisoning ("None" represents the original model). Trends mimic results for clean-label accuracy on unseen samples with the poison trigger.

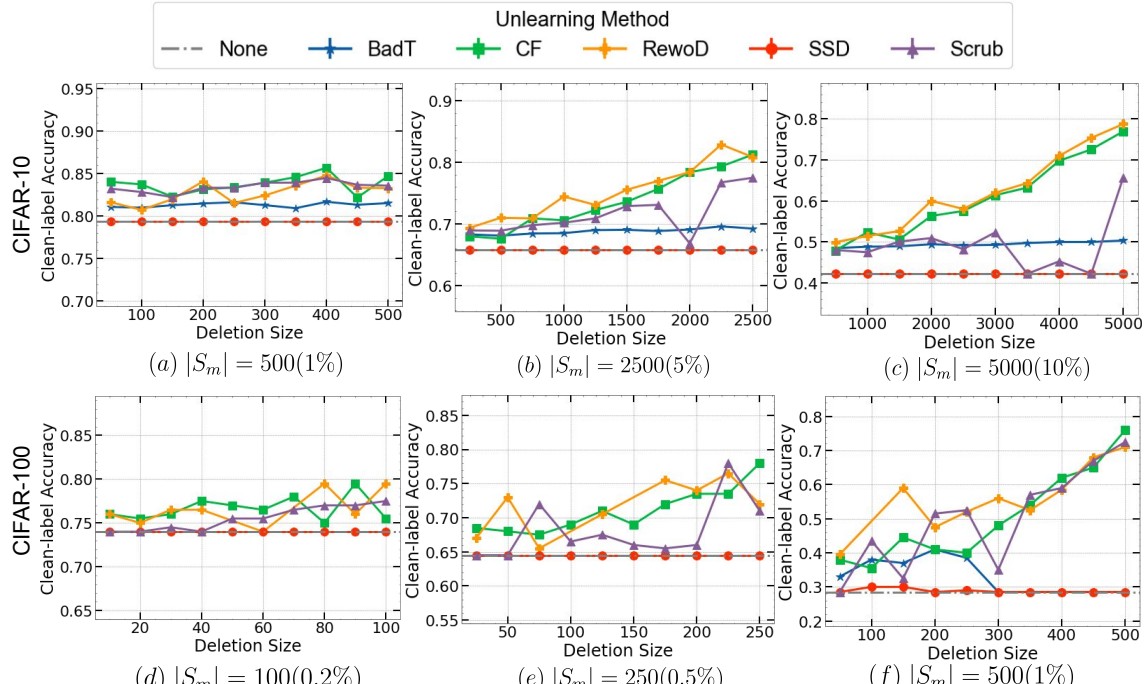

(a) **Clean-label Accuracy on Test Samples on the Two Confused Classes**. We compute clean-label accuracy on the classes $A, B$ used for the Interclass Confusion test, across deletion sizes $|S_f|$. SSD provides no improvements over the original model (represented as "None"), and other unlearning methods also require a large fraction of the manipulated data to be identified for unlearning. In the lower manipulation size setting (a) and (d), the model outputs on unseen samples are not affected much, so we show unlearning trends on manipulated train samples below.

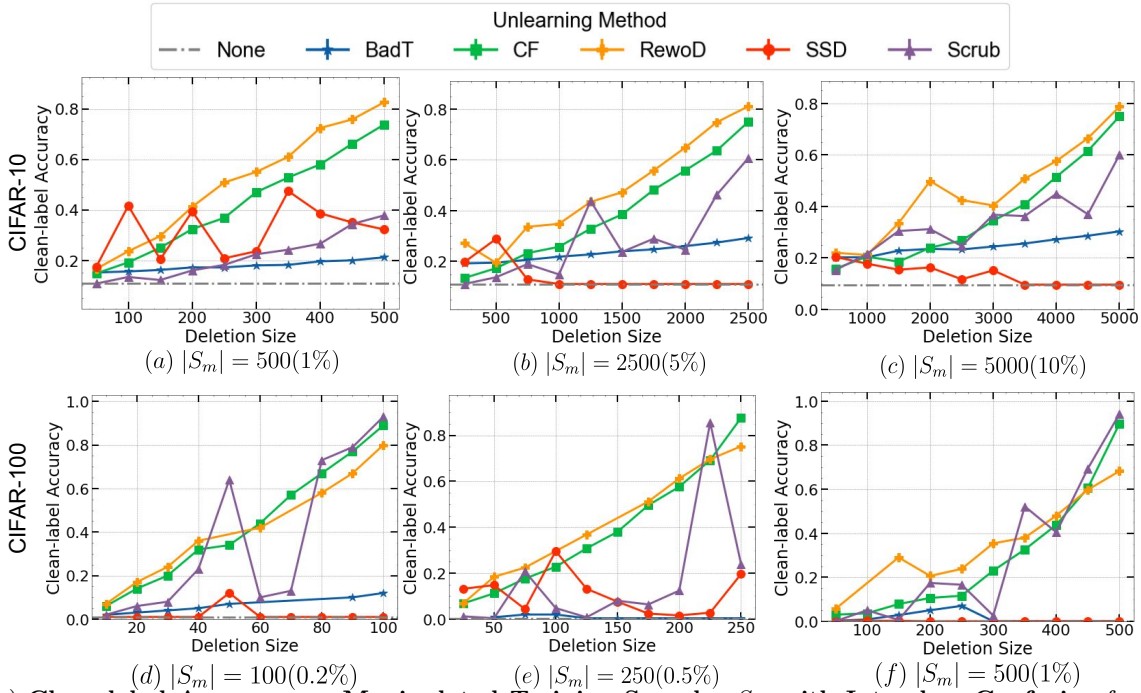

(b) **Clean-label Accuracy on Manipulated Training Samples $S_m$ with Interclass Confusion** for different unlearning methods ("None" represents the original model) across deletion sizes $|S_f|$. Existing unlearning methods perform poorly when $\frac{|S_f|}{S_m}$ is lower. Even the smallest setting (10% of single class) shows clear unlearning trends.

| Method | Time (minutes) |
|--------|------|
| RewoD | 49.93 |
| CF | 10.52 |
| SCRUB | 16.86 |
| SSD | 1.80 |
| BadT | 33.19 |

Table 3: Unlearning Time by Method

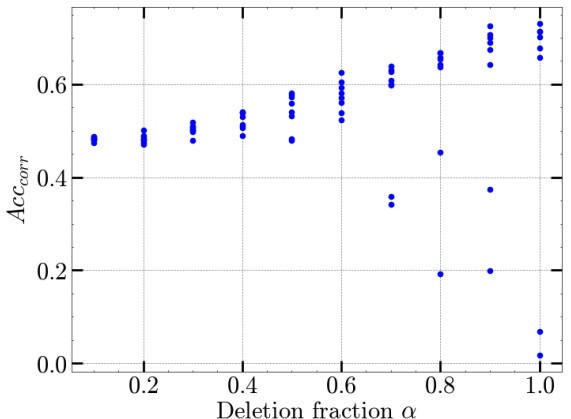

Figure 7: **Unlearning performance of Scrub across hyperparameters on the IC evaluation at different $\alpha$.**

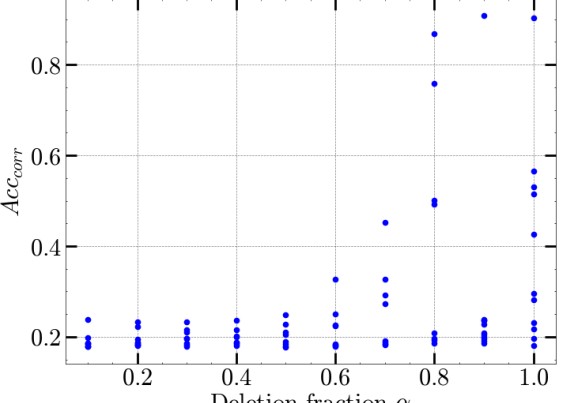

Figure 8: **Unlearning performance of Scrub across hyperparameters on the poisoning evaluation at different $\alpha$.**

### B.3 Hyperparameter Sensitivity

How sensitive are our claims to hyperparameters? For CF, RewoD and BadT, there are no crucial hyperparameters apart from the ones used in the original training procedure, for which we continue to use the same ones as we found them to work quite well. To check hyperparameter sensitivity for Scrub, SSD, we analyze two kinds of plots on the CIFAR10 dataset:

- **Scatter plot of Unlearning across deletion fraction**: We plot the $\mathrm{Acc_{corr}}$ for the model produced by each hyperparameter at values of $\alpha$ from 0.1 to 1 as studied in the main paper.

- **Scatter plot of Unlearning vs Utility at deletion fraction = 0.5**: To compare the unlearning-utility tradeoff, we fix $\alpha = 0.5$ to obtain an intermediate slice. We plot $\mathrm{Acc_{corr}}$ vs $\mathrm{Acc_{retain}}$ for the model produced by each hyperparameter run.

#### B.3.1 Unlearning across deletion fraction

**Scrub**: Figure 7 shows Scrub leads to a gradual improvement in unlearning as $\alpha$ increases, across hyperparameter values. Figure 8 shows that only at high values of $\alpha$, for specific hyperparameter values (high sensitivity), Scrub leads to good unlearning.

**SSD**: Figure 9 shows that SSD is never able to outperform the original model's $\mathrm{Acc_{corr}}$ in the IC evaluation. Figure 10 shows that for specific hyperparameters, SSD performs good unlearning at all values of $\alpha$. The unlearning ability of SSD is highly sensitive to hyperparameters.

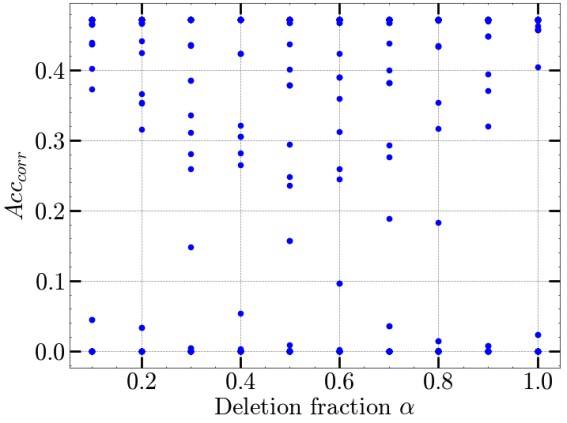

Figure 9: **Unlearning performance of SSD across hyperparameters on the IC evaluation at different $\alpha$.**

Figure 10: **Unlearning performance of SSD across hyperparameters on the poisoning evaluation at different $\alpha$.**

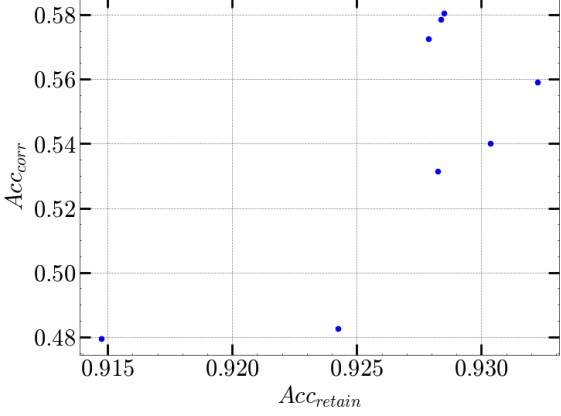

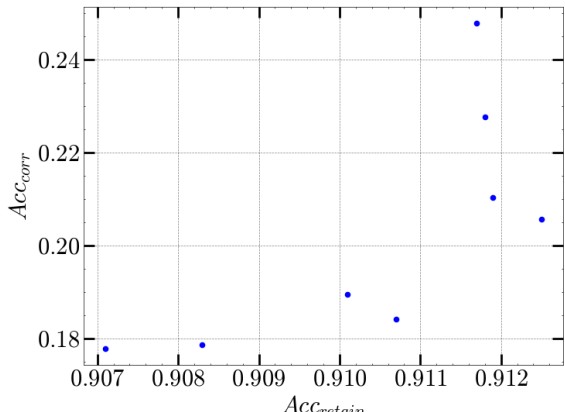

Figure 11: **Unlearning-utility tradeoff of Scrub across hyperparameters on the IC evaluation.**

Figure 12: **Unlearning-utility tradeoff of Scrub across hyperparameters on the poisoning evaluation.**

### B.3.2 Unlearning-Utility Tradeoff

Figures 11, 12 show that Scrub is actually not too sensitive at hyperparameters, and the right values of hyperparameters (for the ones tuned) do not sacrifice unlearning for utility. On the other hand, Figures 13, 14 show that SSD is quite sensitive to hyperparameters. While there is a positive correlation between unlearning and utility, surprisingly, at the highest levels of utility only a few hyperparameter sets lead to high unlearning.

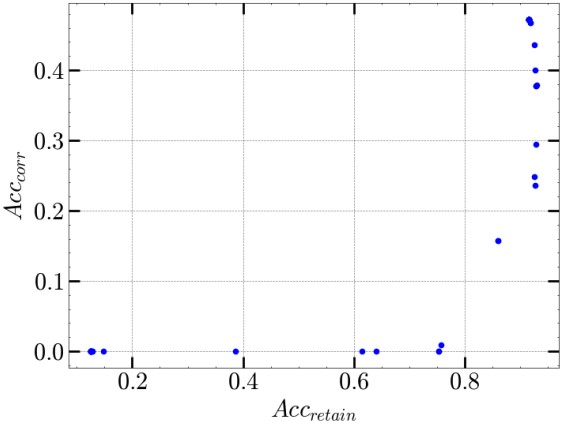

Figure 13: **Unlearning-utility tradeoff of SSD across hyperparameters on the IC evaluation**.

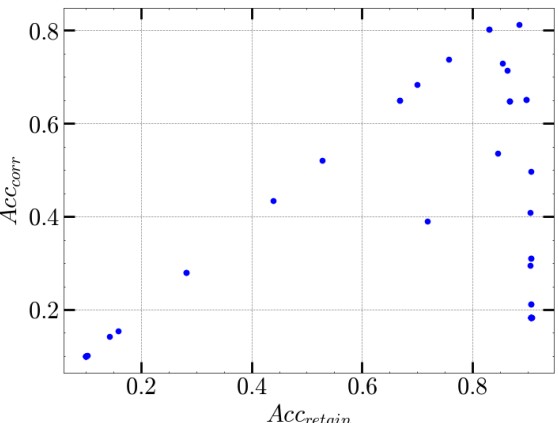

Figure 14: **Unlearning-utility tradeoff of SSD across hyperparameters on the poisoning evaluation**.

