# OpenReview forum: "Corrective Machine Unlearning"
_TMLR — Accepted by TMLR_

### Review · Reviewer_ZM6o · 2024-08-13

**Summary Of Contributions:**

This paper studies a specific use-case for machine unlearning, termed corrective machine unlearning, where the goal is to remove the effect of 'manipulated' training data from a trained model by only having access to a part of the manipulated data, and without any knowledge about the manipulation mechanism. The paper uses two methods to pollute the training data, namely, via a classic poisoning attack, and via a class confusion method. The paper shows that SOTA unlearning algorithm including the gold 'retraining from scratch', is ineffective for such a setting.

**Audience:**

Yes

**Broader Impact Concerns:**

None.

**Claims And Evidence:**

No

**Requested Changes:**

Requested Changes: According to what I discussed above, I think the paper needs to improve/fix the follwoing:


 -- Add the following to the experiments:

 Related baselines

 more datasets

 more model architectures

 Sensitivity analysis of the baselines where applicable

-- report utility using retain accuracy and test accuracy separately.

-- Omit Figure 1 and and split Figure 2.

**Strengths And Weaknesses:**

**Strengths**:

S1. It is very important to investigate different unlearning applications other than privacy. Particularly, data integrity is becoming more important for training ML models and the assumption that developers might only identify a subset of the manipulated data seems reasonable.

S2. The paper is well-written and structured. It is very easy to follow and understand and the experimental details are clear.


**Weaknesses**:

W1. I find the argument around the 'gold unlearning' method inaccurate and misleading. The retrain-from-scratch oracle has a clear definition which is to train a new model only using the remaining data. In your setting, when $\alpha<1$, this baseline is trained on the remaining data plus a part of the forget-set that is not identified. Therefore, it does not have the same definition. So, I don't think it is appropriate to call this 'Exact Unlearning' in your experiments. I think it is important to make this distinction to avoid confusion.

W2. The introduced problem setup in this paper is very difficult. As the results show, only SSD can achieve unlearning in limited cases with small $\alpha$, and even that is with a big drop in utility. This makes me question the claim that this is sufficient evidence to show this problem setup is 'tractable'. To show tractability, more theoretical and empirical evidence is required. To be more concrete, if there exists two poisoned data examples during training that are completely unrelated to each other (for e.g. belong to very different classes), would it ever be possible to unlearn both by only accessing one of them?

W3. The problem formulation in Sec. 2.2, is not accurate. The space of features, $\mathcal{X}$ is defined separatedly from the space of labels, $\mathcal{Y}$. However, $S_m \subseteq S_{tr} \subset \mathcal{X}$ is said to be "training samples manipulated by the adversary, either by modifying features, associated labels, or both."

W4. When discussing model utility, is Eq. (2) expectation over both training examples and test examples? $(x,y) \sim P$ implies that. If that's the case, it would be better to separate the retain accuracy from the test accuracy as they show different information and averaging them hides important facts.

W5. The selected baselines are mainly designed for cases where the entire forget-set is identifiable and accessible. For your problem setup, it would be better to use other baselines that have a more relaxed assumption that make them a better match. Examples are [1, 2, 3]. Also, the model architectures (only resnet family) and the datasets (cifar and PCam) are very limited, making it difficult to make general conclusions.

W6. Regarding the figures: Figure 1 doesn't seem to be very informative or clarifying. The basic setup is clear from the text. it can be omitted. In Figure 2, it is better to present a table separately with a Table caption. Also, it would be better to use the absolute value of the differences and clarify that the magnitude of the difference is the subject and one should look for a lower difference.

W7. It would be a good idea to perform some sensitivity/ablation study of the baselines used, at least in the appendix. For example, SCRUB uses knowledge distillation with an annealing temperature softmax. It would be interesting to see whether increasing the temperature for the max-steps (i.e for the forget-set) would change the results.


[1] Chundawat, V.S., Tarun, A.K., Mandal, M. and Kankanhalli, M., 2023. Zero-shot machine unlearning. IEEE Transactions on Information Forensics and Security, 18, pp.2345-2354.

[2] Chen, R., Yang, J., Xiong, H., Bai, J., Hu, T., Hao, J., Feng, Y., Zhou, J.T., Wu, J. and Liu, Z., 2024. Fast model debias with machine unlearning. Advances in Neural Information Processing Systems, 36.

[3] Chen, Z., Wang, J., Zhuang, J., Reddy, A.G., Silvestri, F., Huang, J., Nag, K., Kuang, K., Ning, X. and Tolomei, G., 2024. Debiasing Machine Unlearning with Counterfactual Examples. arXiv preprint arXiv:2404.15760.

---

> ### Author Response · Authors · 2024-09-05
> **Conceptual Clarifications**
>
> We thank the reviewer for their thoughtful review. We are glad they found the proposed setting important, and the paper well written.
> We begin by clarifying some conceptual issues pointed out by the reviewer, and then provide more empirical evidence to back our claims in the next comment.
>
> **W1**:
> > I don't think it is appropriate to call this 'Exact Unlearning' in your experiments
>
> Thanks for pointing out this potential confusion. We agree it will be better to call it “Retrain Without Deletion set”, now abbreviated RewoD and have updated the paper to reflect this. Earlier we had called it “Exact Unlearning” because it coincides with this gold standard in the traditional $\alpha=1$ setting. We now see that the EU can be confused with the oracle of retraining without the whole manipulated set, which is not known at $\alpha < 1$.
>
> **W2**: Regarding the tractability of the setting, we agree that it is not always possible to unlearn different types of poisons when only a few types are known. This is why we focus on the forget set $(S_f)$ as a "representative subset" of the manipulated set $(S_m)$, as explained in Sections 1 and 2. In Section 2.1, we describe how this representative subset can be obtained through uniform sampling and manual verification of the training data. We believe that when the forget set is indeed representative of the manipulated set, the setting should be tractable. In the case of SSD, it successfully removes the effect of the backdoor, demonstrating tractability however with a drop in overall utility as a side effect.
>
> We do acknowledge more theoretical and empirical characterization of when the forget set $(S_f)$ is sufficient for unlearning the effect of the manipulated set $(S_m)$ is needed. This is something we have already highlighted in our Limitations section.
>
> **W3**: Thanks for pointing out this typo. We meant $S_{m} \subseteq S_{tr} \subseteq \{X \times Y\}$ and have updated this in the paper.
>
> **W4, RC3**: Equation 2 uses the common definition of Population error. $Acc_{retain}$ is a conditional expectation, conditioned on samples not being from the affected subset of the input domain, i.e. $D_m$,
> By “retain”, we did not mean the retained training set but rather the distribution restricted to lie outside $D_m$ (the measure of the training set is $0$ in the distribution $P$) Thus $Acc_{retain}$ is computed on unseen (test) samples not affected by the manipulation, which is the whole clean test set (without trigger) for poisoning, and the remaining classes for IC. The empirical results we report are only over unseen (test) data, and not on the training data. We have now updated Section 3.2 to explicitly mention this, thanks for pointing it out! We understand using the term ‘retain’ can cause this confusion, and thank the reviewer for bringing it to our notice. Would replacing $Acc_{retain}$ with $Acc_{clean}$ throughout the paper alleviate this concern?
>
> **W6, RC4**: We have separated the utility table in Figure 2, and thank the reviewer for helping us improve the presentation.
> > Also, it would be better to use the absolute value of the differences and clarify that the magnitude of the difference is the subject and one should look for a lower difference.
>
> We believe this would be inappropriate in the context of corrective unlearning. While a reduction in utility (negative values in the table) is undesirable, using the absolute value would penalize improvements (positive values) in model accuracy on samples not part of the manipulated domain. In fact, the value should be as positive as possible, though we do not expect accuracy to significantly increase, as seen in the table. We have updated the table caption to mention this.

---

> ### Author Response · Authors · 2024-09-05
> **Evidence**
>
> **W5, RC1**: We agree that the methods we tested were previously studied in a setting where all samples to be removed are known. We believe it is interesting to show that in the practical setting for corrective unlearning, where the whole manipulation set is not specified, these methods don’t work well. It is not clear to us how to apply the methods the reviewer cited in our corrective unlearning evaluations: [1]: The zero shot unlearning method seems to be designed for removing entire classes and it's not trivial to adapt it for our setting. [2, 3]: Both these methods require obtaining ‘counterfactual examples’, and it is not clear to us how to obtain these for poisoned or label confused samples (IC). Thanks for bringing these papers to our notice, we have added this discussion to related work.
>
> As for the extensiveness of our experiments, we would like to note that across {CIFAR10, CIFAR100, PCAM} x {Poisoning, IC} x {3 manipulation sizes in the Appendix} x {Hyperparameter tuning of unlearning methods}, **we ran over 10,000 experiments**. We believe the fundamental idea: *most existing unlearning methods rely on a clean retain set, and fail if manipulated samples that can't be identified remain in it as they reinforce the manipulation*, will not change with architecture, dataset or task. Are there any specific dataset, model settings the reviewer has in mind which would make this core claim more convincing?
>
> **W7, RC2**: We reported the hyperparameter search we performed for all methods in Appendix Section A.2. BadT, CF, EU (now RewoD) do not add hyperparameters apart from the ones used in training (we use the same values).
>
> For Scrub and SSD which have non-trivial hyperparameter choices to be made, **we have now added the following 8 scatter plots to visualize hyperparameter sweeps on CIFAR-10**: {Unlearning vs Deletion Fraction / Unlearning vs Utility} x {Scrub / SSD} x {IC / Poisoning} in Appendix Section B.3. We find SSD is highly sensitive to hyperparameters across $\alpha$, whereas Scrub is more sensitive at higher $\alpha$ which co-incides with when it does good unlearning. Our claims are relatively robust to this as we carefully analyze best models across hyperparameters, keeping in mind the unique constraints this poses in corrective unlearning, as described in Section 3.2. We thank the reviewer for helping improve the evidence presented in our paper.
>
> We thank the reviewer for the suggestion of increasing the temperature for Scrub. We had followed the original work in using the value of 4.0 (in italics below), and focused on tuning $\alpha, \gamma$ as the paper recommended. We report results for Scrub with different temperatures on CIFAR-10 at $\alpha=0.5$ below:
>
> **IC Evaluation**
> | **Temperature** | **$Acc_{retain}$** | **$Acc_{corr}$** |
> | --- | --- | --- |
> | Original Model | 0.923375 | 0.4720 |
> | 0.01 | 0.928375 | 0.5205 |
> | 0.04 | 0.928000 | 0.5415 |
> | 0.10 | 0.929000 | 0.5385 |
> | 0.40 | 0.927625 | 0.5400 |
> | 1.00 | 0.927125 | 0.5385 |
> | *4.00* | *0.925750* | *0.5540* |
> | 10.00 | 0.926250 | 0.5520 |
> | 40.00 | 0.918250 | 0.5815 |
> | 100.00 | 0.926125 | 0.5525 |
> | 400.00 | 0.926875 | 0.5505 |
>
> **Poisoning Evaluation**
>
> | **Temperature** | **$Acc_{retain}$** | **$Acc_{corr}$** |
> | --- | --- | --- |
> | Original Model | 0.9099 | 0.1555 |
> | 0.01 | 0.9148 | 0.1615 |
> | 0.04 | 0.9163 | 0.1737 |
> | 0.10 | 0.9161 | 0.1736 |
> | 0.40 | 0.9158 | 0.1788 |
> | 1.00 | 0.9156 | 0.1746 |
> | *4.00* | *0.9139* | *0.1582* |
> | 10.00 | 0.9131 | 0.1577 |
> | 40.00 | 0.9136 | 0.1580 |
> | 100.00 | 0.9136 | 0.1581 |
> | 400.00 | 0.9136 | 0.1583 |
>
> For IC, unlearning seems slightly better at higher temperatures than lower temperatures as the reviewer expected, but the best value achieved is a modest improvement (58.1 vs 55.4) over our default chosen temperature of 4.0. For poisoning, unlearning seems slightly better at lower temperatures, and once again the best value achieved is only a modest improvement (17.8 vs 15.8) over the default chosen temperature of 4.0. Overall, the main claim, i.e., Scrub performing well only when most of the manipulated set is identified is not sensitive to hyperparameters.

---

> > ### Comment · Reviewer_ZM6o · 2024-09-20
> >
> > I thank the authors for their detailed comments and the new results. Most of my suggestions are applied and some concerns are resolved.
> >
> > I still have one concern (which is major) regarding the tractability of the problem. It shares the same spirit as Reviewer TDvn's first comment.
> >
> > To summarise: the paper introduces a specific problem setup for machine unlearning, and it empirically shows that known unlearning methods fail in this setting. However, the problem setup is inherently very difficult and it breaks the clear assumption of all the studied baselines (that is, to have access to complete D_f). Therefore, it is not unexpected that most of them fail. And, even the successful case of SSD is only achieved by sacrificing utility which makes it useless.
> > I believe introducing a new problem setup (or use case) is only valuable if it is rigorously positioned w.r.t the current state of the arts, or if insights are provided as to how one should approach this problem. None of these is provided in the paper.

---

> ### Author Response · Authors · 2024-09-25
> **Addressing the concern about tractability**
>
> We are happy to know that we have addressed some concerns. Regarding the tractability of the problem, we would like to clarify the following points:
>
> **Tractability**: Our formulation addresses tractability concerns by requiring the deletion set to be a representative sample of the manipulated data, avoiding clear impossibility results. In fact, recent follow up work [1] improves SSD, effectively unlearning the BadNet poison without accuracy drops. This balance between tractability on one manipulation and unresolved challenges for other manipulations makes corrective unlearning a valuable area for future research.
>
> **Real-World Use-Case**: Our problem setup is based on real-world scenarios, as discussed in Sections 1 and 2. In practice, when developers are alerted to manipulated data, only a subset of the data may be identified for deletion. The assumption of knowing the full deletion set has been implicit and taken for granted in current literature, but is not realistic, and we believe highlighting this encourages future research to address this challenge.
>
> **“Expected” Failure of Current Methods**: We believe that probing the limits of current methods is valuable and contributes to a better understanding. For example, we found it surprising that while in IC, methods like SCRUB gradually improve as $\alpha$ increases, in poisoning it is more of a step change at 90%, and methods completely fail for as high as $\alpha=0.8$. This shows existing methods strongly depend on the remaining data being clean -- a new finding.
>
> **SSD**: The strong performance of SSD was surprising with respect to the above point. While some may view the drop in utility as making SSD "useless," the post-unlearning accuracy still remains relatively high (e.g., from 90.52% to 81.58% on CIFAR10 and from 73.79% to 70.54% on CIFAR100).
>
> In summary, we expose an assumption in current unlearning methods that does not hold for real-world unlearning of manipulated data, providing empirical evidence of their failure. We show a path for future work with SSD making partial progress at the loss of some utility (3-9% accuracy). Finally, we also cite follow-up work that has mitigated the accuracy drop of SSD on the poisoning setting, showing the problem we formulated is tractable, while leading to interesting open problems for future work.
>
> [1] Schoepf, Stefan, Jack Foster, and Alexandra Brintrup. "Potion: Towards Poison Unlearning." Journal of Data-Centric Machine Learning Research (DMLR), 2024.

---

### Review · Reviewer_tFZW · 2024-08-22

**Summary Of Contributions:**

The paper considers the problem of post-training mitigation of the negative impacts of identified manipulated training data. To address this problem, the paper takes the perspective of *Machine Unlearning* and refers to the proposed variation as *Corrective Machine Unlearning*. The paper evaluates the effectiveness of different *Machine Unlearning* techniques for this purpose and finds that most are ineffective.

**Audience:**

Yes

**Broader Impact Concerns:**

The paper seeks to mitigate the harmful effects of data poisoning. As such, there are no direct broader impact concerns. There could be secondary impacts where the proposed solution could potentially amplify biases, which the authors acknowledge.

**Claims And Evidence:**

No

**Requested Changes:**

- It is unclear whether the paper is about mitigating the harmful effects of manipulated training data or *Machine Unlearning*. The reviewer acknowledges some similarities to *machine unlearning*. Still, the problem in the paper stands independently (based on differences stated in Section 2.3), and taking a *machine unlearning* perspective feels forceful and unnecessary. Could the authors provide some clarification?
- In the reviewer's opinion the goal of "machine unlearning" is already corrective in nature. It is unclear why "Corrective Machine Unlearning" is the appropriate description for the problem considered in the paper.
- This reviewer is unclear about the paper's precise claims and why they are important. As such, it is challenging to evaluate the claims. Perhaps the claims are implicitly distributed in the paper, but explicitly listing them would improve the paper and allow the reader to assess them.
- The last line of Section 4 suggests that Yoon et al. (2023) is the closest related work. Can you clarify the similarities and differences to that paper and evaluate their approach?
- Make the notation in the problem definition more precise (see minor weakness above).

**Strengths And Weaknesses:**

**Strengths:**

- The paper considers a practically relevant problem: fixing a model trained with manipulated data, where some, but not all, of the manipulated data is known. This scenario is of growing interest, especially as machine learning models are increasingly being trained on massive amounts of web data.
- The paper shows that most of the standard *Machine Unlearning* techniques are ineffective.

**Weaknesses:**


- **Major:** The paper looks at the problem from the perspective of *Machine Unlearning* and hence only evaluates *Machine Unlearning* baselines. There are two issues with this:
    1. It is unclear why one should take a *Machine Unlearning* perspective to the problem. One could take other possible perspectives to mitigate the effects of manipulated data.
    2. Are *Machine Unlearning* baselines the only solution to mitigate the effects of manipulated data? What about other solutions that do not fall under the *Machine Unlearning* umbrella? For example, other methods to mitigate poisoning attacks such as those in the "*Learning from manipulated Data*" part of Section 4.

- **Minor (easily fixable):** In section 2.2, the data domain is $\mathcal{X} \times \mathcal{Y}$. But $S_{tr} \subset \mathcal{X}$ only refers to the data and ignores the labels. Unless the reviewer is missing something, this does not look correct. Shouldn't it include the labels, too, since they are necessary for retraining/fine-tuning? Right now, it reads as if one does not need labels for Corrective Machine Unlearning.

---

> ### Author Response · Authors · 2024-09-05
> **Clarifying the focus on Machine Unlearning**
>
> Thanks for acknowledging that we highlight a practically relevant problem of growing interest. Are responses to the requested changes are as follows:
>
> **W-“Major”, RC1, RC2**: Our paper focuses on what model developers can do when they discover manipulated training data after the model has already been trained. In this scenario, robust learning techniques cannot be applied retroactively. Therefore, we propose unlearning as a post-hoc method to remove the influence of identified manipulated data. This approach complements robust training methods and addresses some of the limitations of those techniques, as mentioned in the Related Work section. We hope this clarifies the motivation behind corrective unlearning.
>
> The traditional machine unlearning literature, as discussed in the ‘Unlearning’ section of Related Work, has primarily focused on privacy concerns—ensuring that information about the data to be deleted is not leaked with different desiderata. Section 2.3 outlines these differences, and for this reason, we titled the paper "Corrective Machine Unlearning" to highlight this distinction.
>
> **RC3**: The main claim of the paper is that the Corrective Unlearning setting has different constraints and goals from the traditional focus of the Unlearning domain on privacy, and thus deserves separate attention. We have now mentioned this explicitly in the introduction, thanks for the suggestion! We first formulate these differences in Section 2.3, and then empirically show that existing privacy-oriented unlearning methods (including the gold standard of retraining without the deletion data) surprisingly fail to work well in the corrective unlearning constraint of not knowing all the manipulated data.
>
> **RC4**: Yoon et al. focus on unlearning from a single class. In contrast, our work formulates the corrective unlearning setting, where unlearning from a representative subset is necessary. Yoon et al.'s method, which generates additional samples for unlearning using a conditional generative model, works well when all the samples come from the same class. However, this approach is not suitable for our poisoning and IC evaluations, where samples come from multiple classes. We attempted to apply their generation method but found it ineffective for generating poisoned and IC samples. While Yoon et al. compare their approach to older unlearning methods, we demonstrate that even recent, widely used methods fail in the corrective unlearning setting. We have added this discussion to the Related Work section. Thank you for pointing this out.
>
> **“Minor” Weakness and RC5**: Thank you for your feedback on the notation. We have corrected it in the revised version of the paper.

---

> > ### Comment · Reviewer_tFZW · 2024-10-03
> > **Appreciation for Addressing Reviewer Concerns**
> >
> > I thank the authors for the rebuttal. It addressed my concerns satisfactorily.

---

### Review · Reviewer_TDvn · 2024-09-01

**Summary Of Contributions:**

This manuscript formalizes a new concept related to a potential application of machine unlearning: addressing data integrity issues. The concept introduced, called "corrective machine unlearning," can be used to mitigate the influence of manipulated data in a pre-trained model, even when only a small portion of the corrupted data is available, and the type and extent of the corruption are unknown. It discusses the differences between corrective machine unlearning and privacy-oriented machine unlearning. Additionally, it experimentally demonstrates that some existing unlearning algorithms perform poorly in scenarios where both labels and features are manipulated or when only labels are corrupted.

**Audience:**

Yes

**Broader Impact Concerns:**

I have no concerns about the ethical implications of this work.

**Claims And Evidence:**

No

**Requested Changes:**

1) Although the authors effectively highlight the problem, they do not propose an initial solution or offer any potential suggestions to address it. Proposing an initial approach to tackle the described problem would significantly enhance the value of this work.

2) On the one hand, the formalized concept of "corrective machine unlearning" is compared with privacy-oriented machine unlearning, highlighting key differences in their goals, applications, and gold standards. On the other hand, some state-of-the-art privacy-oriented machine unlearning algorithms are applied to tackle the corrective machine unlearning problem. The results, unsurprisingly, indicate that these algorithms are ineffective at correcting corrupted data. Given the assumptions made when introducing the concept of "corrective machine unlearning," how would the authors motivate the performed experiments? Prior to designing the experiments,  did they anticipate that existing machine unlearning approaches would be capable of addressing the corrective machine unlearning problem, or were the experiments primarily intended to confirm the differences they highlighted between machine unlearning and corrective machine unlearning?

3) In the experimental evaluation, there is no gold standard to target. The assumed setting is realistic, and a gold standard does not exist. Specifically, there is no approach for correcting the side effects of data manipulation without first identifying all the corrupted data or understanding the type and extent of the manipulation.

4) Moreover, the evaluation metrics require further clarification. The two metrics used for evaluation are corrective accuracy and the average change in retain accuracy over 10 different fractions of identified manipulations (alpha=[0, 0.1, 0.2, .. 1]). However, it is unclear how the latter metric should be interpreted when comparing the performance of the studied approaches and which comparison method is more effective according to this metric. Additionally, the corrective accuracy is the weighted average of the accuracy on a validation set and the accuracy on the identified set of manipulated data. Assuming all the corrupted data is identified (alpha=1), the gold standard would be the exact unlearning algorithm which implies the corrective accuracy will be the average of the validation accuracy and the forget accuracy. However, based on the assumptions made in the “corrective” unlearning problem, the fraction of identified manipulated data is less than 1. The use of this evaluation metric in the absence of a gold standard requires justification, as unknown factors—such as the type of manipulation—can impact this metric.

5) On page 4, section 2.3, in the paragraph titled “Different Goals,” it is stated that “The goal of privacy-oriented unlearning is to remove untampered but sensitive user data...”. This statement can be misleading and requires clarification as the goal of privacy-oriented is removing the influence of sensitive user data, not the data itself.

**Strengths And Weaknesses:**

### Strengths
- Motivating the challenge of correcting the trained model when limited information is available about the manipulation, including its nature, extent, and the specific corrupted data.
- Being well-written and easy-to-follow

### Weaknesses
- Limited contribution
- Limited experimental evaluation

---

> ### Author Response · Authors · 2024-09-05
> **Our contribution is formulating an important setting that has been overlooked**
>
> We thank the reviewer for acknowledging the clear formulation and effective demonstration of the problem of corrective unlearning, which was the goal of this work. Our responses to the feedback are as follows:
>
> **W2, RC2**: Our main claim is that corrective unlearning has distinct constraints and goals compared to privacy-oriented unlearning, as discussed in Section 2.3. We believe it requires separate attention. To support this, we benchmarked popular unlearning methods designed for privacy motivation. These methods do not place specific restrictions on their formulation and claim to handle arbitrary deletion sets $D_f$. Therefore, it was not apriori clear whether these methods would work when the full manipulation set is unknown. For example, it was surprising that SSD performed well with partial identification in a poisoning setting. We did think many existing methods implicitly assume the retain set is clean, which may not be true for corrective unlearning at $\alpha < 1$. For example, we expected Scrub, BadT, CF, and EU, which push towards the original model’s outputs on the retain set, to perform poorly at $\alpha < 1$ as highlighted in Figure 1. Do you have any specific settings in mind where showing more empirical results would make our main claim more convincing?
>
> **W1, RC1**: Having clarified our contribution in the previous point, we do agree proposing a solution would have increased our contribution further. We did some preliminary exploration of ideas for methods, but could not improve unlearning on both the IC and poisoning evaluations simultaneously. We think developing such a method is non-trivial but tractable, and hope that future work can build on our evaluations and achieve this. In a number of other domains, there have been papers focused on identifying a problem.
>
> **RC3**: We acknowledge that there is no gold standard in corrective unlearning, but we do not view this as a weakness. If the full manipulation set is known ($\alpha = 1$), as in privacy-oriented unlearning, the problem becomes purely computational since retraining without the deletion set provides an ideal solution, though an inefficient one. In our setting, the problem also has a statistical dimension—*can we unlearn effectively with only a representative subset of manipulated samples*? We believe it’s valuable to explore how much we can close the gap compared to the oracle retraining without all manipulated data. SSD shows some promise, successfully removing backdoors with just $10%$ of manipulated samples identified.
>
> **RC4**: We measured the average change in retain accuracy across 10 $\alpha$ values to assess whether unlearning methods lead to a drop in  ‘model utility’. We chose to average over these 10 values because we observed similar results across the different $\alpha$ values. Here is the graph of utility across $\alpha$ for the poisoning setting in CIFAR10, CIFAR100 across methods: [Link to graph](https://ibb.co/bL0YBkk). Since there is no significant trend, we believe averaging utility across $\alpha$ does not lose important information.
>
> Regarding corrective accuracy, there seems to be a misunderstanding. In our experiments, corrective accuracy ($Acc_{corr}$ is calculated using unseen test samples, not averaged over validation and manipulated samples. In the definition in Equation 1, the measure of the manipulated set is 0 when computing the expectation. We hope this clears up any confusion, and we have now clarified this in Section 3.2. The confusion may have come from the paragraph on hyperparameter selection, where we averaged the change in manipulated samples and validation accuracy to choose the best model. However, these values are not the ones reported in the empirical results for unlearning and utility.
>
> **RC5**: Thanks for pointing out this typo, we have updated it in the paper.

---

> ### Comment · Reviewer_TDvn · 2024-10-04
> **Response to the Authors' Rebuttal**
>
> I appreciate the authors' efforts in addressing the concerns about the experimental evaluation; however, my main concern still remains unresolved.
> While defining the corrective unlearning problem and demonstrating (both experimentally and in writing) the distinction in constraints and goals of corrective unlearning and privacy-oriented unlearning have educational value, I believe it does not offer generalizable insight.
>
> The assumptions and goals of the defined problem (corrective unlearning) and privacy-oriented machine unlearning are very different. State-of-the-art solutions for privacy-oriented machine unlearning neither promise to mitigate the side effects of a corrupted forget set nor assume partial access to this set. Comparing these two problems and explaining the differences can be educationally valuable, but I’m concerned that it does not provide broader insights for the audience.

---

> ### Author Response · Authors · 2024-10-05
> **Regarding insights and potential impact of our work**
>
> We thank the reviewer for their comment. However, we respectfully disagree with the notion that our work does not offer generalizable insights. We have elaborated in our previous replies the insights contributed by our theoretical formulation backed by extensive experiments into the limitations of existing unlearning methods and real-world use-cases of the corrective unlearning framework. The fact that several follow-up works [1, 2] are able to extend our setting further demonstrates the generalizability of our insights.
>
> Moreover, we believe our work aligns with TMLR's criteria of being relevant to atleast a subset of the audience. The official guidelines explicitly state that projected "impact" and "significance" should not be used as evaluation criteria.
>
> [1] Pawelczyk et al., Machine Unlearning Fails to Remove Data Poisoning Attacks
>
> [2] Schoepf et al., Potion: Towards Poison Unlearning

---

### Decision · Action_Editor_csMX · 2024-10-07

**Recommendation:** Accept with minor revision

**Comment:**

In my view, this paper meets the bar for acceptance to TMLR as it studies an interesting and important topic, and makes a valuable distinction of a relevant and interesting "subproblem" of unlearning from other applications / subproblems, accompanied by a thorough investigation of the suitability and performance of existing unlearning methods for this newly-identified subproblem of "corrective unlearning". As far as I can tell, the claims that the authors make are substantiated by solid evidence, with the potential exception of one relating to tractability, as a reviewer brought up, that I will suggest a minor revision for.

Reviewers found that the paper "considers a practically relevant problem" (Reviewer tFZW), is "well-written and easy-to-follow" (Reviewer TDvn), "well-written and structured" (Reviewer ZM6o) and found that "the experimental details are clear" (Reviewer ZM6o). During the rebuttal, the authors have addressed several reviewer concerns through discussion as well as additional experiments.

While some reviewers found that the experimental results "don't show a significant or novel finding", or felt that some of the conclusions were not surprising, I respectfully disagree with this opinion. I personally (being an expert in unlearning myself) would not have been able to foresee which out of the existing methods would perform best on this type of problem, at what utility cost and under which circumstances. I view it as being very valuable to report on these findings through an empirical investigation. This is important groundwork for future work building on these benchmarks, for instance, and the authors showed during the rebuttal that follow-up work already improves upon prior unlearning methods on their proposed setup. I agree with reviewers that the paper could have been strengthened if a new solution was proposed that worked well for corrective unlearning, but based on TMLR's criteria, the paper in its current form already meets the bar for publication and represents a valuable stand-alone contribution already in my opinion. I partially agree with concerns raised about tractability claims, see below.

Requested minor revisions:

- I agree to some extent with reviewer concerns raised about the claim regarding the "tractability" of the problem. Depending on the relationship ("representativeness") of the identified forget set to the overall manipulated data, I can see variations of this problem that are very challenging, and for which we don't have evidence of tractability as far as I understand (theoretically nor empirically). Can the authors please avoid using the word "tractability", but rather make a more concrete claim, e.g. something along the lines that, for the variations of the problem that are studied in this paper, there is encouraging evidence that progress can be made, even if partially, and discussing concretely which metrics see progress (citing, e.g. the partial success of the SSD approach) rather than making more general claims about tractability. It would be great to also add an explicit note in the limitations section that not all possible instantiations of corrective unlearning may be tractable. This doesn't diminish the relevance of the work (and I agree with the authors that this is a valuable area for future research), but it is important that the claims are as accurate and the relevant nuances are discussed appropriately.

- The authors state that "Our findings indicate that popular unlearning methods, even the traditional gold standard of retraining on the remaining data (RewoD ), fails to enhance accuracy on manipulated domain samples unless nearly all of the manipulated data is identified". I view this phrasing (here and in some other parts of the paper) a little misleading: the gold standard of retraining could in fact also be an ideal solution to this problem (but only if the full set of manipulated samples are identified). Similar to the argument of Reviewer ZM6o for using the term "exact unlearning", it doesn't feel right to refer to retraining on a "retain set" that still contains some manipulated data as a "gold standard". I would rewrite this sentence (and other similar occurrences) to emphasize that i) retrain-from-scratch is the gold standard for the version of the problem where alpha = 1, and ii) for alpha < 1, it should no longer be considered the "gold standard" and may perform worse than other unlearning methods.

**Audience:**

Yes, I find this paper interesting and valuable to TMLR's audience. Establishing a distinction between corrective unlearning from other subproblems of unlearning (e.g. understanding their different characteristics, as well as differences in terms of which methods outshine the others in one setting versus another) is important to guide research into creating new unlearning methods in a way that is grounded in appropriate formulations and goals that capture different types of realistic problems. This is recognized by the reviewers too, e.g. Reviewer ZM6o said "It is very important to investigate different unlearning applications other than privacy".

**Claims And Evidence:**

The authors introduce "corrective unlearning", a subproblem of unlearning where the goal is to remove from a trained model the effect of adversarially manipulated training data of which only a (representative) subset is identified, and only post training. The authors claim that this variant of unlearning is fundamentally different from "unlearning for privacy", and they claim that unlearning methods developed for different formulations of unlearning do not work well in corrective unlearning. They claim that the problem is challenging but "tractable".

The claim that existing unlearning methods developed for different applications don't work well in this setting is substantiated through a thorough empirical investigation across datasets, unlearning methods, percentage of identified manipulated data, as well as different types of manipulations (e.g. "The paper shows that most of the standard Machine Unlearning techniques are ineffective" - Reviewer tFZW). They find that only one prior method works "well" (but at some utility cost) and only for one type of adversarial manipulation they consider. They view this as evidence that the setting is "tractable", since it's possible to make progress, even if partially (and follow-up work that the authors cite in the rebuttal shows even further improvement, strengthening this argument). I agree with concerns raised by reviewers about the tractability claim (see below Comment) and suggest a minor revision to address this.

---

> ### Author Response · Authors · 2024-10-15
> **Thank you for the feedback, we have fixed the issues in the camera ready**
>
> We thank the Action Editor and reviewers for their detailed feedback which has helped improve our work.
>
> In the camera ready version we uploaded, we have changed uses of the word 'tractability' to make more concrete claims. We have also added an explicit note in the limitation that corrective unlearning may not always be tractable. We have also rephrased our claims related to retraining from scratch. We hope this sufficiently incorporates the suggested revisions.

---

> > ### Comment · Action_Editor_csMX · 2024-10-21
> > **thank you**
> >
> > Dear authors, thank you for making these changes, the adjusted draft looks good to me!